# Evolution Pattern in Bruised Tissue of ‘*Red Delicious*’ Apple

**DOI:** 10.3390/foods13040602

**Published:** 2024-02-16

**Authors:** Tao Xu, Xiaomin Zhang, Yihang Zhu, Xufeng Xu, Xiuqin Rao

**Affiliations:** 1College of Biosystems Engineering and Food Science, Zhejiang University, Hangzhou 310058, China; 2Key Laboratory of on Site Processing Equipment for Agricultural Products, Ministry of Agriculture, Hangzhou 310058, China; 3Key Laboratory of Intelligent Equipment and Robotics for Agriculture of Zhejiang Province, Hangzhou 310058, China; 4National Key Laboratory of Agricultural Equipment Technology, Hangzhou 310058, China

**Keywords:** mechanical damage, cell death zone, evolution pattern

## Abstract

The study of apple damage mechanisms is key to improving post-harvest apple treatment techniques, and the evolution pattern of damaged tissue is fundamental to the study of apple damage mechanisms. In the study, ‘*Red Delicious*’ apples were used to explore the relationship between damage and time. A cell death zone was found in the pulp of the damaged tissue after the apple had been bruised. The tissue damage was centered in the cell death zone and developed laterally, with the width of the damage increasing with injury time. The extent of tissue damage in the core and pericarpal directions varied. About 60% of the damaged tissue developed in the core direction and 40% in the pericarpal direction, and the damage ratios in both directions remained consistent throughout the injury. The depth of damage and the rate of damage were influenced by the impact force size and the difference in the size of the damaged part of the apple, but the damage development pattern was independent of the impact force size and the difference in the damaged part. The maximum damage rate was reached at about 30 min, and the depth of damage was stabilized at about 72 min. By studying the evolution pattern of the damaged tissue of the bruised ‘Red Delicious’ apple, it provides the research idea and theoretical basis for enhancing the prediction accuracy and robustness of early stage damage in apples.

## 1. Introduction

Mechanical damage from falls, collisions, squeezing, vibrations, etc., is one of the main causes of fruit loss during post-harvest treatment of apples. The deterioration of apples due to mechanical damage and increased losses due to the infection of rotten apples result in severe losses to producers and consumers. After mechanical damage, apples are prone to microbial infection and rapid reproduction at the damaged site, accelerating decay and deterioration [1], thus affecting their edible value and economic benefits. According to statistics, the mechanical damage rate of Chinese apples is as high as 25–30% [2]. This loss can be reduced and fruit quality can be improved if injury identification and removal can be achieved during the early stages of mechanical damage. Scholars have conducted exploratory studies of early damage detection by using spectroscopic and imaging techniques, but the accuracy and robustness of the detection are far from meeting the requirements. The study of the mechanism of apple damage at the tissue level is key to the problem and provides an important theoretical basis for early apple injury identification techniques.

After mechanical damage occurs, the apple cells and tissue structures are damaged. The apple skin is composed of epidermal cells and closely packed subcutaneous villi tissue (about six layers of villi cells), through which impact forces are transmitted to the flesh [3]. Pulp is mainly composed of parenchyma cells with intercellular space, which is vulnerable to external damage [4,5]. Both living cells and dead cells that are broken, crushed or without obvious damage can be found in the flesh tissue of the injured pulp [6]. However, different varieties of apples showed different states of damaged tissue after a collision. According to Harker et al. [7], tissues have three failure modes: cell rupture, cell division, and intercellular disadhesion. Cell rupture is the breaking of cells at the equator, cell division refers to cell fragmentation and collapse, and intercellular disadhesion refers to intercellular separation. Ng et al. [8] observed the fracture surface of ‘*Scifresh*’ and ‘*Royal Gala*’ fruits using scanning electron microscopy, and found significant differences between the two fracture surfaces. Li et al. [9] found that the pulp of ‘*Fuji*’ and ‘*Ruiyang*’ was mainly composed of broken cells, while the pulp of ‘*Qinguan*’ was comprised mainly of broken cells and intercellular disadhesion, and thus speculated: The flesh of the varieties with brittle flesh was dominated by cell rupture, while the flesh of the varieties with loose flesh was dominated by cell division and intercellular disadhesion. Seymour et al. [10] believed that the reason for the intercellular disadhesion might be a strong cell wall in pulp or a weak intercellular connection. Intercellular space was the weakest point in the tissue, and the increase in intercellular space due to intercellular disadhesion greatly aggravated the degree of cell rupture near the space [6,11]. Alvarez et al. [4] believed that the larger the cell, the larger the cellular space, and the smaller the adhesion of adjacent cells. The above studies, which mainly focus on the failure modes of the apple tissue after the collision, still lack an exploration of the variation patterns of the damaged tissue in the temporal dimension and are not able to formulate a perfect damage mechanism for the apple. As a consequence, there is a need to carry out research on the form of apple damage and the expansion process of damaged tissue to lay the foundation for a dynamic model of damage volume throughout the entire process of damage.

The author’s team analyzed the relationship between the morphology of the early internal damaged tissues and the distribution of impact energy in ‘*Red Delicious*’ apples, established the relationship between damage parameters and the force of damage, proposed the morphological characteristic of ‘cell death zone’ as a feature of apple damage tissue, and completed the preliminary study of the morphology of damaged apple tissues [12]. The aim of this study is to analyze the extent of internal tissue damage after impact, establish the relationship between the extent of damage and time, and directly and comprehensively reflect the evolution of damage on a spatiotemporal level using ‘*Red Delicious*’ apples as the research subject and the cell death zone as the basis. This provides a research direction and theoretical basis for further improving the prediction accuracy of the severity of injuries.

## 2. Materials and Methods

### 2.1. Research Materials and Instruments

The same batch of ‘*Red Delicious*’ apples was purchased from the local wholesale fruit market for the experimental material. All the apples selected were of similar size, weight, hardness, ripeness, and free from cracks, rot, and other injuries. A total of 200 apples were randomly selected and measured one by one for diameter, height and weight [12,13]. Due to the conical shape of the ‘*Red Delicious*’ apple (fruit shape index 0.9–1.0) and its distinct pentagonal features, we conducted diameter measurements at the upper, middle, and lower sections along the central axis. For the upper section diameter measurement, the fruit was rotated by 120° for each of the 3 measurements, and then the average of these measurements was taken as the diameter of the upper section. Similarly, measurements were taken and averaged for the middle and lower sections of the fruit. The average diameter of the top, equator and bottom of the apple was 73.26 mm, 63.27 mm and 65.01 mm, respectively, and the average height was 87.69 mm. The mean weight was 223.35 g [13,14,15].

A testbed, shown in Figure 1, was developed for the experiment of apple forming. An impact test apparatus (model MY-A721, Dongguan Mingyu Intelligent Technology Co., Ltd., Dongguan, China) was composed of a base, ruler, guide rod, buckle and impact hammer. The impact hammer had a diameter of 80 mm and a weight of 500 g [12].

Equipped with a 20-million-pixel color microscopic digital imaging system (model MC-DK20U3(C)Pro, Phoenix Optics Group Co., Ltd., Shangrao, China), a stereo-microscope (model XTL-165-MT with field view diameter 4.8 mm–31.5 mm, Phoenix Optics Group Co., Ltd., Jiangxi, China) was used for microscopic image acquisition.

### 2.2. Damage Observation and Positioning Method

Due to the thick skin and dark color of the ‘*Red Delicious*’ apple (bright red or thick red) (Figure 2A), it was hard to observe the damage. However, the aqueous solution of vitamin B2 produced a distinct fluorescence in the ultraviolet band, and it did not penetrate to the interior of the apple which would have affected the follow-up experiment. Therefore, 10% of the aqueous solution of vitamin B2 was rubbed onto the impact surface of the sensor. Upon impact, the aqueous solution of vitamin B2 adhered to the surface of the apple. The damaged location could be observed under the ultraviolet lamp (395 nm) (Figure 2B) [12].

### 2.3. Experiment of Mechanical Damage

The upper, waist, and lower parts of the apple were chosen as the impact surfaces, respectively, and the apple was securely placed on the fruit cup. The aqueous solution of 10% vitamin B2 was rubbed onto the impact surface of the hammer. The ruler slider of the impact test pedestal was moved to drive the impact hammer to 125 mm. After the buckle was pressed down, the impact hammer was dropped free to hit the apple in the fruit cup to complete the apple forming. Twenty samples were replicated at each site [12].

The upper part of the apple was selected as the impact region after the sample was stabilized in a fruit cup. Therefore, 10% of the aqueous solution of vitamin B2 was rubbed onto the impact surface of the hammer. The ruler slider of the impact test apparatus was moved to drive the impact hammer at the specified height. After the buckle was pressed, the impact hammer fell free and the apple was struck on the specified area of the sample. The impact hammer was released at a series of specified heights (50 mm, 75 mm, 100 mm, 125 mm, 150 mm, 175 mm, 200 mm, respectively) to cause damages on the sample’s surface. A sample of 20 apples was taken for each height.

### 2.4. Collection of Damage Image

The injured parts were determined under ultraviolet lamp (395 nm). The central point of the impact position and the plane located at the axis wire of the apple were used as the segmentation plane (Figure 3). The sections of the injured parts were regarded as objects to make tissue sections. The sections were cleaned with phosphate-buffered saline (PBS), and then stained with 0.04% trypan blue solution at room temperature for about 30 s. After dyeing, they were washed with PBS buffer again [16]. Tissue profile images were collected and spliced under a microscope, in combination with a color microscopic digital imaging system [12].

Apple tissue slices were stained and images were collected every 6 min. Each apple was continuously collected 30 times (180 min) to form a microscopic image group of continuous changes of the damaged site 180 min after the injury, named T_1_, T_2_,… T_30_ and C_1_, C_2_,… C_30_.

### 2.5. Extraction of the Cell Death Zone

Due to the large flesh cells and brittle flesh of the ‘*Red Delicious*’ apple, a tissue fracture surface exists at a certain depth under the skin of the damaged site after impact (the three-dimensional tissue fracture surface presents as a band at the angle of tissue slice and the phenomenon of concentrated cell death exists in this fracture zone, hereinafter referred to as a cell death zone). The cell death zone occurred when the impact force caused the cell to break apart, creating a fracture zone around the dying cell that eventually formed a cell death zone [13].

The cell death zone, which was composed of concentrated dead cells, could also be caused by cutting during tissue segmentation. The dead cells were blue after staining with 0.04% Trypan blue solution (Figure 4) [12]. These dead cells were divided into three categories. The first was the cell death zone formed by injury. The second was spread over the observed surface of the tissue profile, showing a roughly uniform distribution over a small range of clustering. The third was distributed on the sides and bottom of the tissue section, concentrated and resembling the shape of a cell death zone.

The color difference in the stained cell death zone was poor compared to the rest of the apple tissue slice, with the main difference being the different color saturation. Therefore, after separating HSV color space channels and graying, S (saturation) channels were selected to extract the damage area (Figure 4) [12].

Since the purpose of the experiment is to observe the evolution pattern of the injured tissue, the development of the second type of dead cells will affect the observation of the evolution of the cell death zone after the introduction of the time dimension; therefore, it is necessary to modify the damage gray map of the same tissue section at different times. Then, the undamaged tissue part of the T_30_ grayscale of the same tissue section (Figure 4) was selected as the benchmark to calculate the differences of the mean gray value in the gray image of T_2_, T_3_,…, T_30_ and T_1_. After subtracting the differences, the grayscale images of T_2_, T_3_,…, T_30_ was approximated to correct the effect caused by the second type of dead cells.

The binarization operation, morphological treatment, extraction of maximally connected components, and hole filling were performed on the corrected grayscale image, as shown in Figure 4. The middle 1/3 part of the image was processed using the smallest outer rectangle, and a line was taken at the middle position of the two rectangles for target segmentation. By obtaining the average width of the cell death zone, the intersection between the two ends of the cell death zone and the two sides of the tissue section was corrected (Figure 4).

### 2.6. Collection of Damage Parameterthe

The two-dimensional coordinates of the apple surface and the edge of the cell death zone in the component diagram of S-channel were collected using Matlab R2022a (MathWorks. Inc, Natick, MA, USA), and the data such as the distance from the cell death zone to the skin surface and the width of the cell death zone (hereafter referred to as the depth and width of the cell death zone) could be obtained by simple calculation on the Y-axis. However, when taking data in the Y-axis direction, discontinuities in the cell death zone were encountered. The midpoint of the two furthest points was chosen as the central position of the cell death zone for the depth calculation, and the sum of the widths of all the cell death zones in this direction was chosen as the width. Taking the situation in Figure 5 as an example, (Y_1_ + Y_4_)/2 was selected as the central position of the cell death zone for the calculation of the depth, and (Y_2_ − Y_1_) + (Y_4_ − Y_3_) was selected as the width.

### 2.7. Extraction of Damage Tissues and Parameter Collection

The following three sets of microscopic images with continuously varying of damage sites were chosen. Groups T_1_, T_2_, …, T_30_ and C_1_, C_2_, …, C_30_ were used as the data to carry out follow-up studies, including: (1) The upper part of the apple was used as the impact surface to impact 20 apple samples with a height of 125 mm, which was used to study the relationship between the damaged tissue and time; (2) The upper, the waist and the lower parts of the apple were used as the impact surface, respectively. A total of 60 apple samples with a height of 125 mm were impacted to study the relationship between tissue damage and time at different injury sites; (3) The upper part of the apple was used as the impact surface, and 140 apple samples with impact heights of 50 mm, 75 mm, 100 mm, 125 mm, 150 mm, 175 mm and 200 mm were used to study the relationship between damaged tissue and time with different damage degrees.

The above mentioned Section 2.5 and Section 2.6 were referred to the extraction and parameter acquisition processes of damaged tissues, where the difference lied in the modification process induced by the effect of the second type of dead cells. The second group of dead cells actually consisted of dead cells from the surge in levels of hormones like ethylene. The dead cells in the preparation of tissue sections were interference factors introduced in the experimental observation process and need to be distinguished. Therefore, the modification process affecting the second type of dead cells in Section 2.5 needed to be modified.

The non-marginal part of the C_30_ grayscale image of the same tissue slice was chosen as a reference for calculating the difference of the average graying values in the grayscale image of C_2_, C_3_, …, C_30_ and C_1_, respectively named d_2_, d_3_, … d_30_. The grayscale images of T_2_, T_3_, …, T_30_, subtracted the d_2_, d_3_, …, d_30_, of the corrected grayscale images, which were considered to modify the influence of cell death during tissue section preparation.

## 3. Results and Discussion

### 3.1. Analysis of Evolution Pattern of the Cell Death Zone

The upper part of the apple was chosen as the impact surface, and the cell death zone of 20 apple samples with a height of 125 mm was impacted to analyze the depth and width continuum variability data.

#### 3.1.1. Temporal Characteristics of the Width of the Cell Death Zone

The cell death zone in the ‘*Red Delicious*’ apple occurred because the impact force of a blow caused the cell to break apart, creating a fracture zone around the cell, which eventually formed a cell death zone. As a result, the continuous cell death around the fault zone led to an increase in the width. In other words, changes in the width might reflect cell death around the fault zone. As shown in Figure 6, the width of the cell death zone increased with time for post-impact damage. Considering the relationship between the depth of the cell death zone and time in Section 3.1.2, the depth of the zone increased slightly with the passage of time of damage. The analysis showed that during the development of the width, the development rate of cell death was different on both sides of the fracture zone, and the development rate was faster near the fruit core. The results showed that the width of the cell death zone increased with time of injury (coefficient of determination R^2^ was 0.866, *p* < 0.001) and then stabilized.

#### 3.1.2. Temporal Characteristics of the Depth of the Cell Death Zone

As shown in Figure 7, the depth of the cell death zone is almost independent of the time of damage. In particular, the depth of the cell death zone increased slightly with time due to the fact that the depth data were obtained based on the width of the center. In the process of changing the width of the cell death zone the two sides of the death scenario developed at different rates, with the side closer to the core developing faster resulting in a slight drift in the data. The results showed that the depth of the cell death zone did not change with the time of injury (coefficient of determination R^2^ was 0.211, *p* < 0.05). The cell death zone is deeply determined by the location of the fracture zone formed by the physical difference between the impact force and the damage site of the ‘*Red Delicious*’ apples.

### 3.2. Analysis of the Evolution Pattern of Damaged Tissue

The causes of death of dead cells in apples after impact injury were as follows: (1) impact directly led to cell rupture, resulting in cell necrosis [6]; (2) intercellular fracture was caused by impact, and cells on both sides of the fracture zone lost protection, resulting in cell necrosis [6,9]; (3) ethylene and other hormones were increased during injury development [17,18,19], causing premature cell maturation, senescence and death [20,21]. The first two types of dead cells were already included in observational studies of the cell death zone, while the third type, due to its relatively homogeneous distribution, was modified to account for dead cells caused by tissue slicing during the extraction of the cell death zone. The causes of cell death were different, but the outcome was the same during the development of tissue damage. From this, it was possible to study the evolution patterns of the damaged tissue.

The upper part of the apple was selected as the impact surface, and data on the persistent changes in the damaged tissue of 20 apple samples with impact heights of 125 mm were analyzed. As shown in Figure 8, the damage was centered on the cell death zone and developed to both sides. The damage width increased with the passage of time and stabilized after about 72 min (core direction: coefficient of determination R^2^ was 0.988, *p* < 0.001; pericarp direction: determination coefficient R^2^ was 0.957, *p* < 0.001). At this impact height, the depth of the damage increased by about 7 mm after the development trend of the damage site stabilized. The degree of the development of tissue damage was different in the direction of the core and in the direction of the pericarp. The development of the damage in the direction of the core was about 4.2 mm, accounting for 60%, and the development in the direction of the pericarp was about 2.8 mm, accounting for 40%. The ratio of damage to the two directions fluctuated in time dimension, and the ratio of damage to the core was 55–64% (Figure 9). There was a limit to the development of damaged tissue in the direction of the pericarp (it cannot exceed the pericarp), and considering the proportion of tissue damage development in the direction of the pericarp, the depth of the cell death zone determined the development limit of damaged tissue in the direction of the pericarp to a certain extent. Therefore, it was concluded that the development limit of the damaged tissue, namely the depth of damage, was affected by the impact force and the physiological difference in the damaged site of the ‘*Red Delicious*’ apple, because the cell death zone was deeply affected by the impact force and the physiological difference in the damaged site of the ‘*Red Delicious*’ apple.

The impact height was 125 mm, and the impact surface was 20 apple samples from the upper, waist, and lower parts. As shown in Figure 10, due to the anisotropy of the apple, the damage depth of the damaged tissue in the upper, waist and lower parts of the ‘*Red Delicious*’ apple were slightly different, with the maximum damage depth in the waist and the minimum damage depth in the lower part. Through first-order derivation of the damage depth curve (Figure 10), the damage rate curve (Figure 11) was obtained. The progression of damage at the three sites was consistent (Pearson correlation coefficient between the two was greater than 0.990). After impact, the depth of damage increased with time of damage, reaching the maximum damage rate around 30 min, and the development of damage began to level off around 72 min (Figure 11).

The upper part of the apple was selected as the impact surface, and data on the persistent changes in the damaged tissue were analyzed for 20 apple samples with impact heights of 50 mm, 75 mm, 100 mm, 125 mm, 150 mm, 175 mm, and 200 mm. As shown in Figure 12 and Figure 13, the damage depth at the same site increased with impact height. The progression of damage sites was consistent at different impact heights (Pearson correlation coefficient between the two was greater than 0.968). The damage depth increased with the passage of time, reaching the maximum damage rate at about 30 min and leveling off at about 72 min.

Experimental results showed that both the damage depth and the damage rate of the ‘*Red Delicious*’ apple were affected by the impact force and the difference in the damage site. The progression of the damage was independent of the impact force and the difference in the damage sites. The maximum damage rate was reached at about 30 min, and the damage depth stabilized at about 72 min. Ebrahim et al. [22] found that there was a temperature difference between the damaged and undamaged parts of an apple after bruising, with the temperature difference reaching a maximum at about 30 min. The reason was that the water content in the damaged area changed during the development of the damaged parts, and the maximum damage rate led to the maximum temperature difference.

## 4. Conclusions

In the pulp of the damaged site, formed by the impact of the ‘*Red Delicious*’ apple, there was a banded tissue fracture surface with concentrated cell death, namely the cell death zone. The evolution pattern is as follows:

The width of the cell death zone increased with the time of injury (R^2^ = 0.866, *p* < 0.001), and then stabilized. During the development of the width, the rate of cell death varied between the two sides of the fracture zone, with a faster rate near the core. The depth of the cell death zone did not change with the time of injury (R^2^ = 0.211, *p* < 0.05), and its value was affected by impact force and physiological difference in injured sites of the ‘*Red Delicious*’ apple.

The tissue damage was centered on the cell death zone and developed to the sides of the core and pericarp. The damage width increased with the observation time and stabilized after 72 min. The development degree of the cell death zone in the direction of core and pericarp was different. Damage to the tissue in the direction of the core accounted for about 60% of the damage, while damage to the tissue in the direction of the pericarp accounted for about 40%. The proportion of damaged tissue in the direction of the core fluctuated in the temporal dimension and ranged from 55% to 64%. The depth of tissue damage was affected by the impact force and physiological differences in the injury site of the ‘*Red Delicious*’ apple.

Due to the anisotropy of apples, when the impact height was the same, the damage depth of the damaged tissue at the upper, waist and lower parts of the ‘*Red Delicious*’ apple were slightly different, with the maximum damage depth at the waist and the minimum damage depth at the lower part, but the damage development pattern of the three parts was consistent. After impact, the damage depth increased with time, reaching the maximum damage rate at about 30 min, and beginning to stabilize at about 72 min. The damage depth of the same site increased with the increase in impact height, and the development pattern of the damage site was consistent at different impact heights. The damage depth increased with the observation time, reaching a maximum damage rate at about 30 min and leveling off at about 72 min.

## Figures and Tables

**Figure 1 foods-13-00602-f001:**
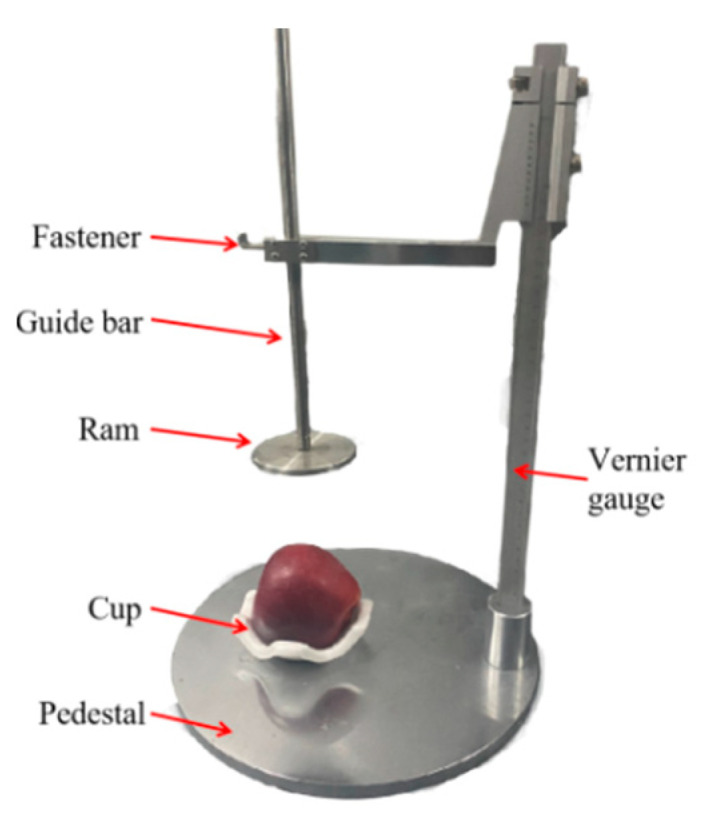
The testbed of apple impact test.

**Figure 2 foods-13-00602-f002:**
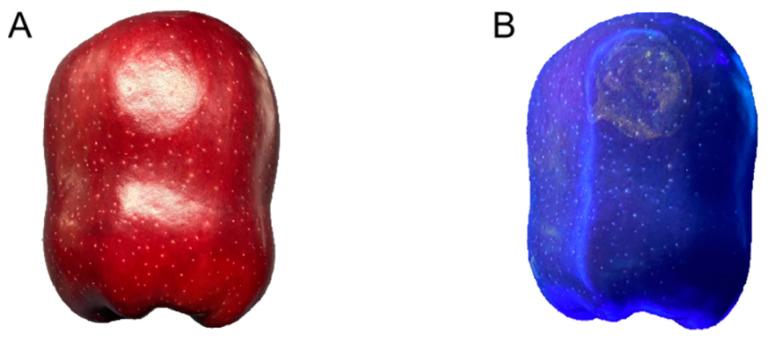
The injured parts of the ‘*Red Delicious*’ apple under different lights. (**A**) Sunshine; (**B**) Ultraviolet.

**Figure 3 foods-13-00602-f003:**
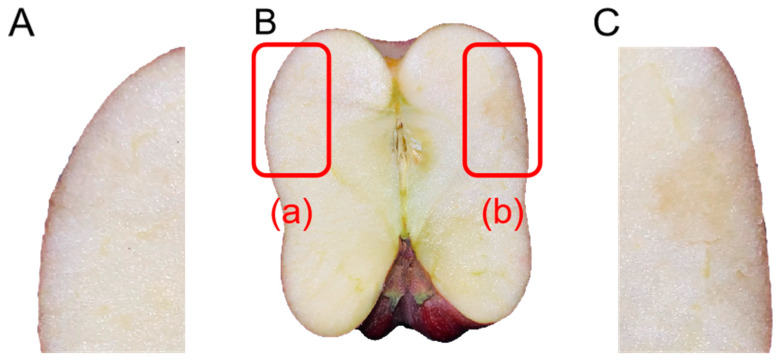
The injured parts sectional drawing of the ‘*Red Delicious*’ apple. (**A**) The axisymmetric position of the injured site; (**B**) Apple profile; (**C**) Site of damage. The red square (a) represents area A in the figure, and the red square (b) represents area B in the figure.

**Figure 4 foods-13-00602-f004:**
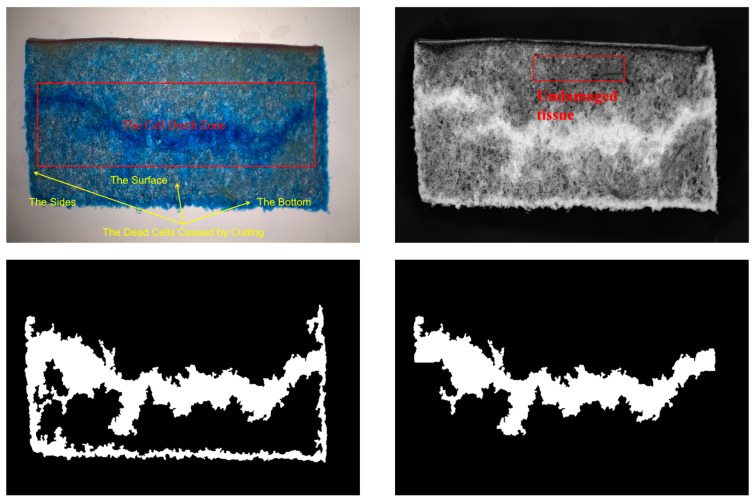
Extraction process of the cell death zone feature.

**Figure 5 foods-13-00602-f005:**
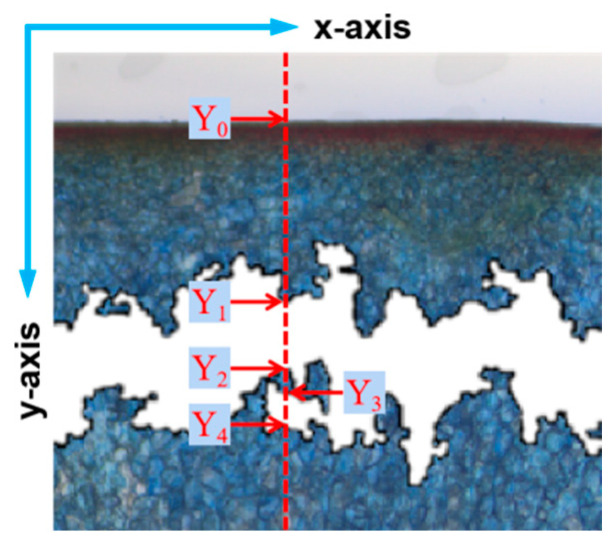
The 2D coordinates of the cell death zone.

**Figure 6 foods-13-00602-f006:**
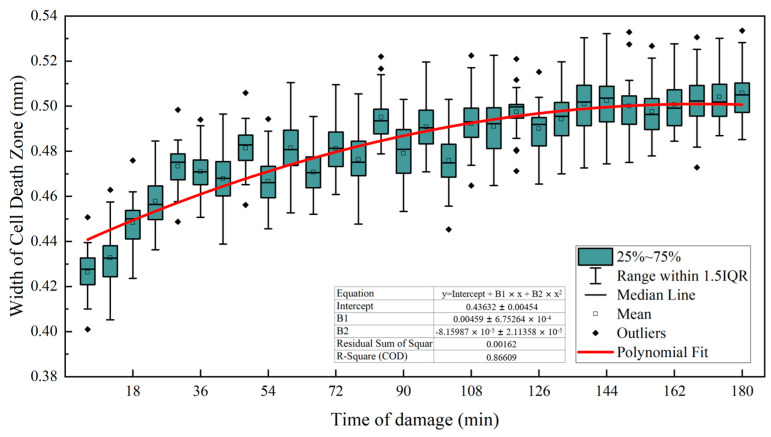
Relationship between the width of the cell death zone and time of damage.

**Figure 7 foods-13-00602-f007:**
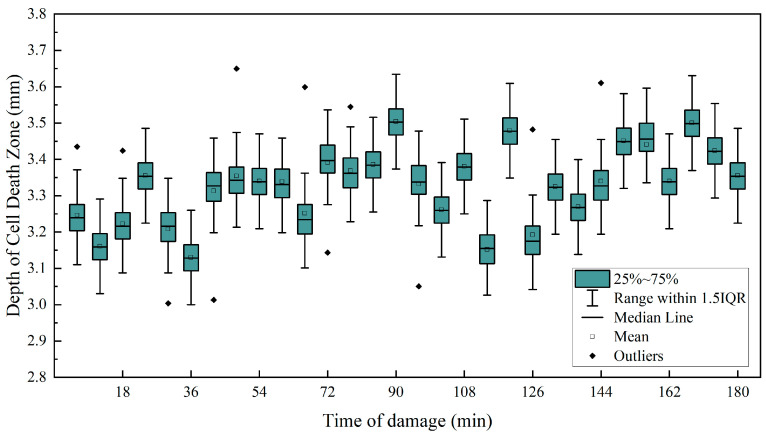
Relationship between the depth of the cell death zone and time of damage.

**Figure 8 foods-13-00602-f008:**
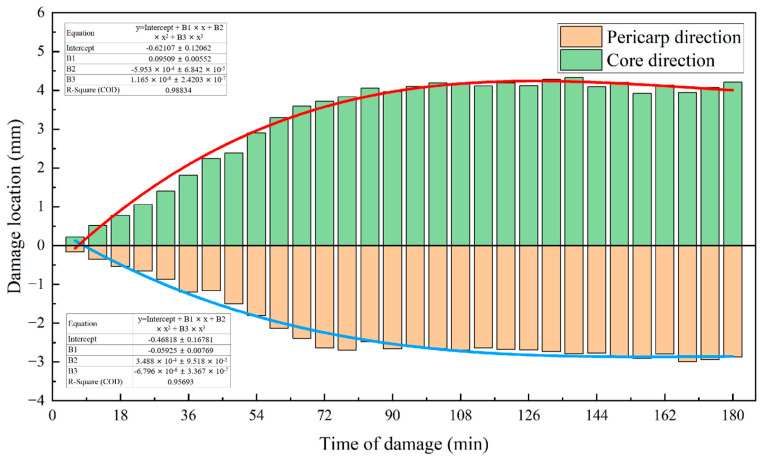
Relationship between the damage location and time of damage. The core direction is positive. The red line is the fitted curve of the damage position in the core direction, and the blue line is the fitted curve of the damage position in the pericarp direction.

**Figure 9 foods-13-00602-f009:**
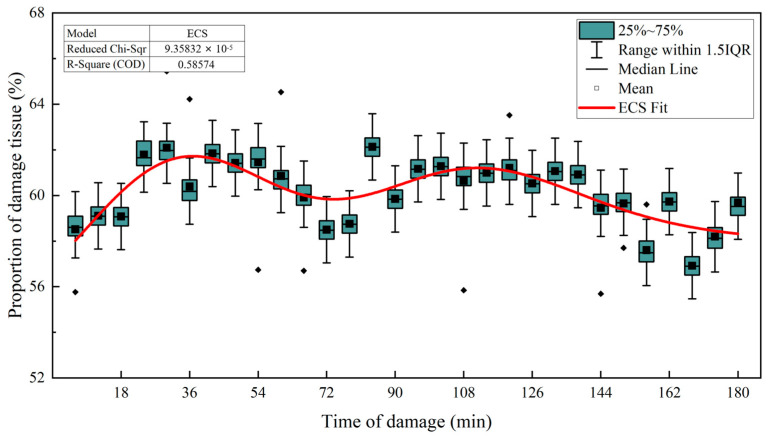
Relationship between the proportion of damaged tissue and time of damage.

**Figure 10 foods-13-00602-f010:**
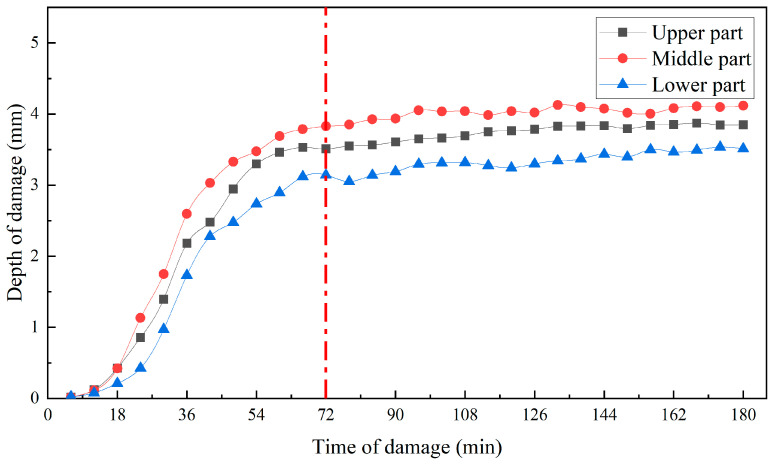
Relationship between the damage depth of the different damage locations and time of damage. The development of damage began to level off around 72 min.

**Figure 11 foods-13-00602-f011:**
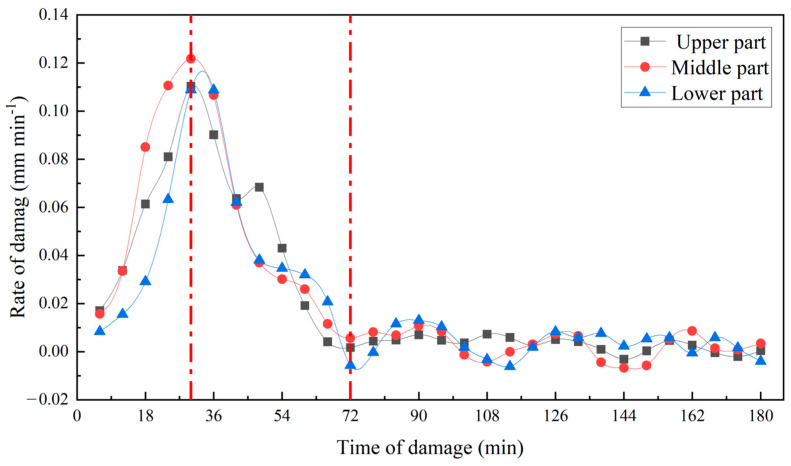
Relationship between the damage rate of the different damage locations and time of damage. The depth of damage increased with time of damage, reaching the maximum damage rate around 30 min, and the development of damage began to level off around 72 min.

**Figure 12 foods-13-00602-f012:**
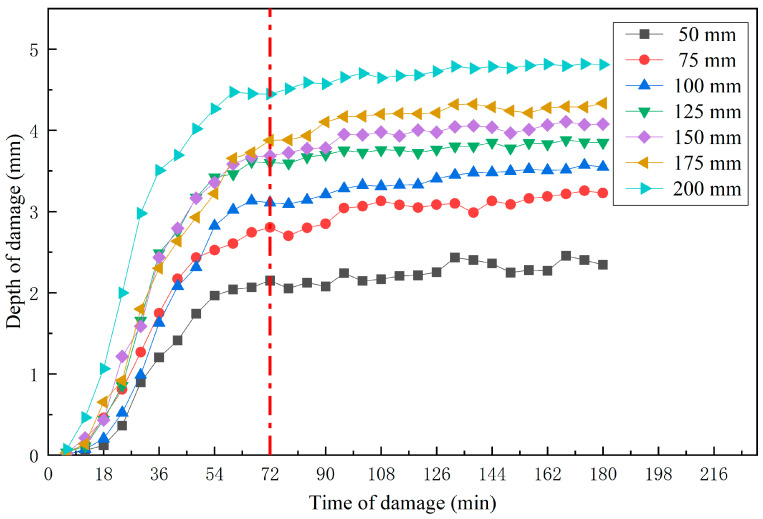
Relationship between the damage depth of the different impact height and time of damage. The development of damage began to level off around 72 min.

**Figure 13 foods-13-00602-f013:**
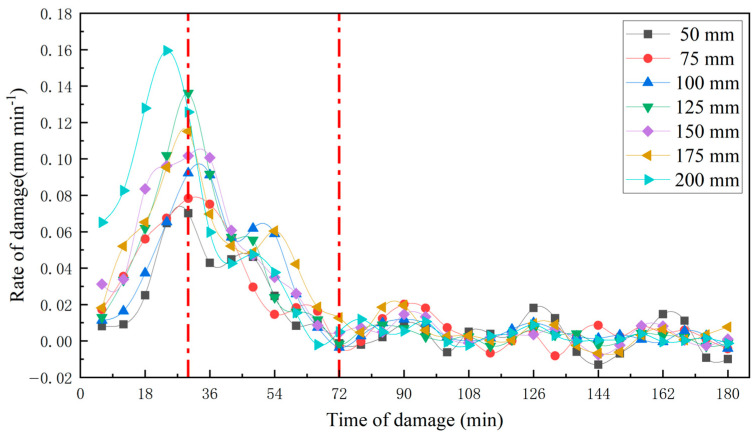
Relationship between the damage rate of the different impact height and time of damage. The depth of damage increased with time of damage, reaching the maximum damage rate around 30 min, and the development of damage began to level off around 72 min.

## Data Availability

The data presented in this study are available on request from the corresponding author. The data are not publicly available because they are intended for use in other ongoing research and should be protected before official publication.

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
