# Peer review of "Evolution Pattern in Bruised Tissue of ‘Red Delicious’ Apple"

_foods, 2024, doi:10.3390/foods13040602_

Round 1
Reviewer 1 Report
Comments and Suggestions for Authors
General: The topic is interesting and has scientific and practical significance. But there are some inaccuracies and errors.
1. Style should be improved. Repeating the same words in a sentence should be avoided. For example:
Mechanical damage from falls, collisions, squeezing, vibrations, etc. is one of the 29 main causes of fruit loss during post-harvest treatment of apples. Deterioration of apples 30 due to mechanical damage and increased losses due to the infection of rotten apples will result in severe losses to producers and consumers. This loss can be reduced and fruit quality can be improved if damage detection and removal can be achieved during the early stages of mechanical damage. Scholars have conducted exploratory studies of early damage detection by using spectroscopic and imaging techniques, but the accuracy and robustness of the detection are far from meeting the requirements. The study of the mechanism of apple damage at the tissue level is key to the problem and provides an important 37 theoretical basis for early apple damage detection techniques.
After mechanical damage occurs, the apple cells and tissue structures are damaged.
2. Please, check the style of references in the text of the manuscript.
3. It will be useful to expand Introduction and include (a) economic assessment of damage harvested crops; (b) methods used for protection of fruits and vegetables from post harvested damage.
4. Since the research topic has applied knowledge, it should be reflected in the statement of purpose. Otherwise, it is not clear how the results of the study can be used as an example.
5. Lines 39-45. It will be useful to present an illustration for this description.
6. It is unclear what the authors are referring to in this case: “The same batch of 'Red Delicious' apples were purchased from the local wholesale 80 fruit market for the experimental material (Wang et al., 2022; Xu et al., 2023)”.
7. Figures 1 and 2 should be joined in one.
8. Figure 3. Image of an undamaged fresh apple may be added.
9. Capture to Figure 6. “Correlation between the depth of the cell death zone and time of damage”. Could you please explain what the expression “time of damage” means? Is it the time after the damage has occurred, or the storage time of the damaged apple?
10. In conclusion it should be explained what practical meaning this research has.
11. References should be edited according to the instructions.
It is a pity that authors do not pay attention to new methods to prolong shelf life of fruits and vegetables.
Pirog, T., Stabnikov, V. & Stabnikova, O. (2022). Bacterial microbial surface-active substances in the food-processing industry. In O. Paredes-López, O. Shevchenko, V. Stabnikov, & V. Ivanov, (Eds.), Bioenhancement and fortification of foods for a healthy diet (pp. 271-294). CRC Press, Boca Raton, London. DOI: 10.1201/9781003225287-18
Reviewer 2 Report
Comments and Suggestions for Authors
Revision: Foods - Evolution Pattern in Bruised Tissue of 'Red Delicious' Apple by Xu et al.
Authors tried to elucidate the mechanism of apple damaging to improve post-harvest treatments. Actually the study is not conclusive as they also say that is the “ basis for a subsequent, more reliable study”. Why this was not reliable?
There are several typos and in general the English language needs a revision.
INTRODUCTION
I think that a histological explanation of apple fruit anatomy and histology would be helpful.
MM
Line 80: was instead of were?
Lines 87 89, please rephrase.
Line 107-108 I think these considerations are not appropriate to a research paper.
Paragraph 2.2 is not clear, suddenly vitamin B2 is mentioned without explaining anything. I think an explanation can fit also in the introduction.
Paragraph 2.4 how did authors obtain sections? How many sections were observed?
Something important to point out is that the authors use the word “kernel” to mean maybe the fruit core!? But, according to the histo anatomy of fruits, this is correct only for drupes, not for the apple. Apple do not have kernel. They should completely revise apple anatomy.
Very minor changes in the attached PDF.

There are several typos and grammatical mistakes to correct.
Reviewer 3 Report
Comments and Suggestions for Authors
Manuscripts Foods-2831873
The manuscript deals with the relationship between damage and time, using “Red Delicious” apples. The experimental design is simple but easy to interpret, and the results are modest. However, information on the evolution of damages after picking bruising is scarce. The text is clear and concise. However, this reviewer considers that the use of correlations in the legends could be misleading. They show simply the changes with time. Also, the software used for the study, graphs, etc., is missing. Finally, some graphs lack information on data variability; justification of absence or inclusion is necessary
L 185. Please, correct
L 337-340. It is unclear the procedure and the parameters used to study the relationships between the width and depth of cell death with time. This aspect requires clarification.
Comments on the Quality of English LanguageIn my opinion, correct English text
Round 2
Reviewer 3 Report
Comments and Suggestions for Authors
Manuscript improved. No further comments